

# Genomics enters the deep learning era

Etienne Routhier[1] and Julien Mozziconacci[2]

[1] LPTMC, Sorbonne Université, Paris, France
[2] StrInG Lab, Museum National d'Histoire Naturelle, Paris, France

## ABSTRACT

The tremendous amount of biological sequence data available, combined with the recent methodological breakthrough in deep learning in domains such as computer vision or natural language processing, is leading today to the transformation of bioinformatics through the emergence of deep genomics, the application of deep learning to genomic sequences. We review here the new applications that the use of deep learning enables in the field, focusing on three aspects: the functional annotation of genomes, the sequence determinants of the genome functions and the possibility to write synthetic genomic sequences.

## INTRODUCTION

Genomics is the field in life science focusing on genomic sequences (Fig. 1A) and attempting to link the DNA sequence of a living organism with its physical and molecular characteristics. High-throughput sequencing techniques provide huge amounts of data to reconstruct this link. These techniques can now provide both the linear genome sequence and a lot of other information such as the genome 3D structure in cells (Hi-C), the nucleosome and other proteins bindings sites found along the molecule (MNase-seq, ChIP-seq), the local accessibility of the DNA sequence (DNase-seq), the epigenetic marks found on nucleosomes (ChIP-seq) and the activity of genes (RNA-seq, CAGE). Machine learning has long played an important role in the processing of these huge amounts of data (*Libbrecht & Noble, 2015*) and deep learning has recently emerged as a promising methodology to renew these machine learning approaches. This trend is shared by all bio-medical fields (Fig. 1A) for which the number of publications regarding the application of deep learning is exploding (*Holder, Haque & Skinner, 2017*; *Ho et al., 2019*; *Zitnik et al., 2019*; *Zemouri, Zerhouni & Racoceanu, 2019*).

Schematically, machine or deep learning has been applied to genomics for two main tasks (Fig. 1B). First, it has been used to predict second order (*i.e.* functional) annotation using the first order annotation (*i.e.* the experimental measures such as ChIP-seq, RNA-seq…). This process consists in labeling each DNA segment along the genome with a function (*e.g.* promoter, protein binding site, enhancer…). We will call here this general task *genome annotation*, going beyond the mere annotation of genes. Second, machine learning can also be used to annotate (first and/or second order) the genome directly from

Corresponding authors
Etienne Routhier,
etiennerouthier@gmail.com
Julien Mozziconacci,
julien.mozziconacci@mnhn.fr

**(a)**

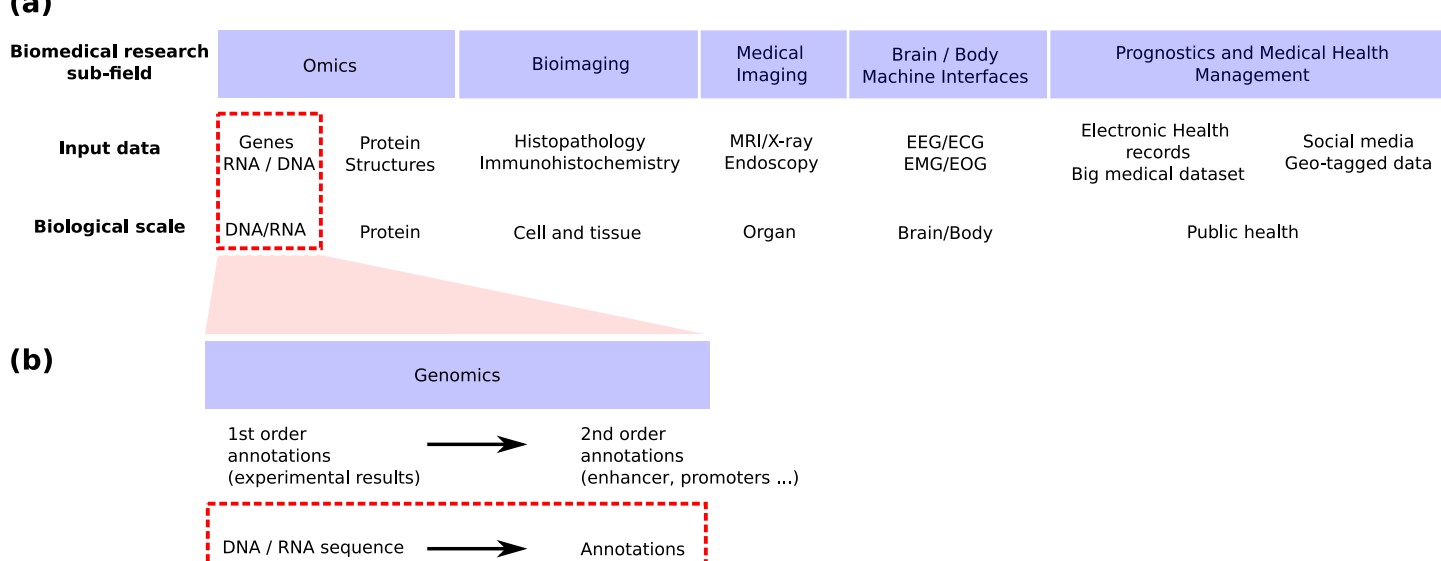

**(b)**

Figure 1 **Positioning of this review within the field of deep learning for biological/biomedical application.** (A) We adapted the segmentation of the field proposed by *Zemouri, Zerhouni & Racoceanu (2019)* to position this review. (B) Zoom into the genomics field. This review focus on the application of deep learning to annotate the genome directly from the DNA sequence (red dashed line boxes).

the DNA sequence. This review focuses on the application of deep learning for this second learning for this second task (Figs. 1A and 1B, red dotted line boxes).

Other reviews focus on the application of deep learning to genomics and proteomics (*Zou et al., 2019*; *Eraslan et al., 2019*; *Zhang et al., 2019c*; *Yue & Wang, 2018*), often with an introduction to technical aspects and a rather broad domain focus. They present the different neural network architectures used in various types of applications as well as the potential pitfalls (*Koumakis, 2020*). Our goal here is to provide a complementary view, focusing on the practical benefits of the application of deep learning to the task of genome annotation from the DNA sequence. This review is intended to biologists and bioinformaticians who are curious to know what new questions can be efficiently tackled using deep learning, how deep learning may help them in their own studies and maybe change their perspectives on their field.

Amongst the first and most emblematic methodologies that were proposed, DeepSEA (*Zhou & Troyanskaya, 2015*) DeepBind (*Alipanahi et al., 2015*) and Basset (*Kelley, Snoek & Rinn, 2016*), are similar in both technical set up and goals. They all use a Convolutional Neural Network (CNN) architecture, which was originally used in computer vision (*Fukushima, 1980*; *LeCun et al., 1989*) to predict whether an input sequence is accessible (*i.e.* harbors a DNAse peak), contains transcription factors binding sites (TFBS), or specific histone modifications (as assessed by ChIP-seq). Since these pioneering approaches were developed, the number of different methodologies used in the field has rapidly grown and the diversity in both domain of application and technical methods has exploded (see Table 1).

We divide the application of deep learning to genome annotation into three different goals (Fig. 2): (1) transferring a known annotation for a given species or a given cell type to

**Table 1** Overview of studies applying deep learning in genomics, segmented by their usage.

| Annotation | Usage | Preprocessing | Data | Species | Architecture | Reference |
|---|---|---|---|---|---|---|
| TFBS | Transfer | one-hot-encoding | DNA + gene expression + DNaseI cleavage | human | CNN + RNN | *Quang & Xie (2019)* |
| | | | DNA sequence | human + mouse | CNN | *Cochran et al. (2021)* |
| | Bio. mechanism | one-hot-encoding | DNA sequence | human | CNN | *Wang et al. (2018b)* |
| | | | | human + mouse + drosophilia | CNN | *Wang et al. (2018a)* |
| | | | RNA sequence | human | CNN | *Koo et al. (2018)* |
| | Syn. genomics | one-hot-encoding | DNA sequence | human | RNN + Attention | *Gupta & Kundaje (2019)* |
| | | | | | CNN | *Lanchantin et al. (2016)* |
| TFBS + histone + chromatin accessibility | Transfer | one-hot-encoding | DNA sequence | human + mouse | CNN | *Kelley (2020)* |
| | Bio. mechanism | one-hot-encoding | DNA sequence | human | CNN | *Kelley et al. (2018)* |
| | | | | | CNN | *Alipanahi et al. (2015)* |
| | | | | | | *Zhou et al. (2019)* |
| | | | | | | *Hoffman et al. (2019)* |
| | | | | | | *Zhou & Troyanskaya (2015)* |
| | | | | | | *Richter et al. (2020)* |
| | Syn. genomics | one-hot-encoding | DNA sequence | human | CNN | *Schreiber, Lu & Noble (2020)* |
| TFBS (circRNA) | Bio. mechanism | one-hot-encoding | RNA sequence | human | CNN | *Wang, Lei & Wu, 2019* |
| chromatin | Transfer + Bio. mechanism | one-hot-encoding | DNA + gene expression | human | CNN | *Nair et al. (2019)* |
| accessibility | Bio. mechanism | one-hot-encoding + embedding | DNA sequence | human | CNN | *Liu et al. (2018)* |
| gene expression | Transfer + Bio. mechanism | one-hot-encoding | DNA + TF expression level | yeast | CNN | *Liu et al. (2019)* |
| | Bio. mechanism | one-hot-encoding | RNA sequence | 7 species | CNN | *Zrimec et al. (2020)* |
| | Syn. genomics | | | yeast | CNN | *Cuperus et al. (2017)* |
| | | | DNA sequence | Random promoters (yeast) | CNN + Attention + RNN | *Vaishnav et al. (2021)* |
| | Bio. mechanism | one-hot-encoding | DNA + mRNA half-life + CG content + ORF length | human | CNN | *Agarwal & Shendure (2020)* |
| | | | DNA + promoter-enhancer interaction | human | CNN | *Zeng, Wang & Jiang, 2020* |
| | | | DNA sequence | human | CNN | *Movva et al. (2019)* |
| gene expression + RNA splicing | Syn. genomics | one-hot-encoding | DNA sequence | human | CNN | *Linder et al. (2020)* |

**Note:**
CNN, convolutional neural network; RNN, recurrent neural network. After the pioneering use of CNN in genomics in 2015, the methodologies have diversified according to four different aspects: the model inputs (that may include other annotations on top of the sole DNA sequence), the sequence encoding (mainly one-hot-encoding or k-mer embedding), the neural network architecture (CNN, RNN, Attention mechanism) and the output format, which can be either binary or continuous.
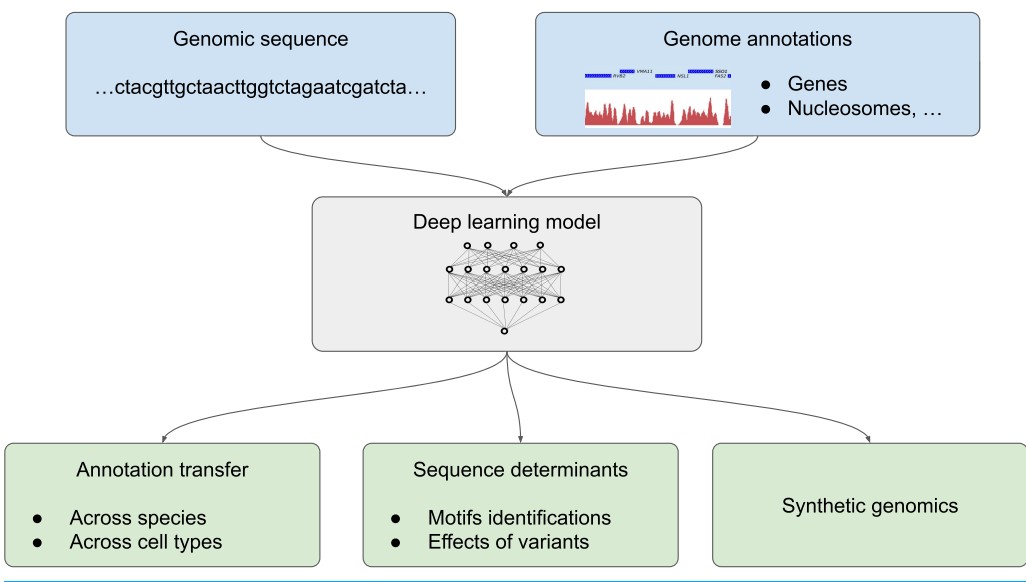

**Figure 2 Different possible uses of deep learning in genomics.** Deep learning models trained with genome annotations together with the underlying genomic sequence (in light blue) can be used for thre different applications (in light green) (1) to automatically annotate the genome of a given species and for a given cell type, (2) to determine the sequence determinants of the genome functions by identifying sequence motifs (such as position weight matrix, PWM) and the effect of sequence variants, or (3) to design artificial sequences.

another species or a different cell type, practically enabling the automatic annotation of genomes from the DNA sequence; (2) getting a deeper understanding of sequence determinants of the genome function by predicting the effect of non-coding mutations and determining sequence motifs that are recognized by the cell machinery; (3) designing synthetic DNA sequences with a tailored annotations (Fig. 2). Two related applications for which classification methodologies from machine learning are inherently useful are the tasks of species attribution for a given, usually short, sequence and the task of assessing sequencing errors are covered in the insert "Sequence classification".

For the sake of completeness and for those who are interested in more technical aspects, we also compiled a short description of the publications that focus on methodological advances (Table 2).

## SURVEY METHODOLOGY

Deep learning for genomics is a rapidly evolving field. We did our best to gather many of the studies published so far focusing on the use of deep neural networks that take as input genomic sequences. To ensure an unbiased analysis of literature, a comprehensive analysis of published articles was carried using the following online databases: Medline (PubMed), Science Direct (http://sciencedirect.com) database, and Google Scholar database. We used the following keywords: deep learning, neural networks, genomics, DNA sequence and then gathered articles together with articles that were cited within the recovered articles.
**Table 2 Overview of studies developing deep learning methodologies in genomics.**

| Annotation | Usage | Preprocessing | Data | Species | Architecture | Reference |
|---|---|---|---|---|---|---|
| epigenetic mark | Bio. mechanism | one-hot-encoding | DNA + chromatin accessibility | human | CNN | *Yin et al. (2019)* |
| | | | DNA + CpG neighborhood of cells | human | CNN + RNN | *Angermueller et al. (2017)* |
| | | | DNA sequence | human | CNN | *Zeng & Gifford (2017)* |
| | | | RNA sequence | human + mouse + zebrafish | CNN + RNN | *Zhang & Hamada (2018)* |
| polyadenylation | Bio. mechanism | one-hot-encoding | DNA sequence | *Arabidopsis thaliana* | CNN | *Gao et al. (2018)* |
| | | | | human | CNN | *Leung, Delong & Frey (2018)* |
| | Syn. genomics | one-hot-encoding | DNA sequence | human | CNN | *Bogard et al. (2019)* |
| polyadenylation + translation initiation site | transfer | one-hot-encoding | DNA sequence | human + mouse + bovine + drosophilia | CNN | *Kalkatawi et al. (2019)* |
| splicing | Bio. mechanism | one-hot-encoding | DNA sequence | human | CNN | *Cheng et al. (2019)* |
| | | | | | | *Cheng et al. (2021)* |
| | | | | | | *Du et al. (2018)* |
| | | | RNA sequence | human | CNN | *Jaganathan et al. (2019)* |
| | | | | | CNN | *Wang & Wang (2019)* |
| D architecture | Bio. mechanism | one-hot-encoding | DNA sequence | human | CNN | *Zhou (2021)* |
| | | | | | CNN | *Fudenberg, Kelley & Pollard, 2020* |
| | | | | | CNN + RNN | *Singh et al. (2019)* |
| | | | | human + mouse | CNN | *Schwessinger et al. (2020)* |
| nucleosome | Bio. mechanism | one-hot-encoding | DNA sequence | yeast | CNN | *Routhier et al. (2021)* |
| nucleosome + TFBS | Bio. mechanism | one-hot-encoding | DNA sequence | yeast + human | CNN | *Cakiroglu et al. (2021)* |
| enhancer | transfer | embedding | DNA sequence | 6 species | CNN | *Chen, Fish & Capra (2018)* |
| | Bio. mechanism + Syn. genomics | one-hot-encoding | DNA sequence | 6 pecies | CNN + RNN | *Minnoye et al. (2020)* |
| | Bio. mechanism | one-hot-encoding | DNA sequence | human | CNN | *Min et al. (2017b)*) |
| promoter | transfer | one-hot-encoding | DNA sequence | 5 species | CNN | *Khodabandelou, Routhier & Mozziconacci (2020)* |
| promoter + enhancer + TFBS + chromatin accessibility | Bio. mechanism | one-hot-encoding | DNA sequence | human | CNN | *Wesolowska-Andersen et al. (2020)* |
| translation initiation site | Bio. mechanism | one-hot-encoding | DNA sequence | human | CNN + RNN | *Zhang et al. (2017)* |
| sgRNA | Syn. genomics | one-hot-encoding | DNA + TFBS + epigenetic + accessibility | human | CNN | *Chuai et al. (2018)* |
| binding site | | | DNA sequence | human + mouse | CNN | *Xue et al. (2018)* |

*(Continued)*

| Annotation | Usage | Preprocessing | Data | Species | Architecture | Reference |
|---|---|---|---|---|---|---|
| Virus integration | Bio. mechanism | one-hot-encoding | DNA sequence | human | CNN + Attention | *Tian et al. (2021)* |

Overview of studies applying deep learning in genomics, segmented by their usage.

| Annotation | Preprocessing | Data | Species | Architecture | Reference |
|---|---|---|---|---|---|
| TFBS | benchmark | DNA sequence | human | benchmark | *Trabelsi, Chaabane & Ben-Hur, 2019* |
| | embedding | DNA sequence | human | CNN + RNN | *Zhang, Shen & Huang (2019a)* |
| | one-hot-encoding | DNA + distance to various annotations | human | CNN | *Avsec et al. (2018)* |
| | | DNA + histone marks + accessibility | human | CNN | *Jing et al. (2019)* |
| | | DNA + shape | human | CNN | *Zhang, Shen & Huang (2019b)* |
| | | DNA sequence | human | CNN | *Brown & Lunter (2019)* |
| | | | | CNN + RNN | *Zhang et al. (2020)* |
| | | | | CNN | *Shrikumar, Greenside & Kundaje (2017b)* |
| | | | | CNN | *Luo et al. (2020)* |
| | | | | CNN | *Zeng et al. (2016)* |
| | | | | CNN | *Chen, Jacob & Mairal, 2019* |
| TFBS + histone marks + accessibility | one-hot-encoding | DNA sequence | human | CNN | *Cao & Zhang (2019)* |
| | | | | | *Kelley, Snoek & Rinn, 2016* |
| | | | | | *Tayara & Chong (2019)* |
| | | | | CNN + RNN | *Quang & Xie (2016)* |
| | | | | CNN + Attention | *Avsec et al. (2021a)* |
| | | | | CNN | *Gupta & Rush (2017)* |
| chromatin accessibility | embedding | DNA sequence | human | CNN + RNN | *Min et al. (2017a)* |
| epigenetic marks | embedding | RNA sequence | human | CNN | *Mostavi, Salekin & Huang, 2018* |
| polyadenylation | embedding | DNA sequence | 4 species | CNN + Attention | *Guo et al. (2021)* |
| | one-hot-encoding | RNA + secondary structure | human | CNN + RNN | *Arefeen, Xiao & Jiang, 2019* |
| 3D architecture | one-hot-encoding | DNA + DNAseI signal | human | CNN | *Schreiber et al. (2017)* |
| nucleosome | one-hot-encoding | DNA sequence | yeast | CNN | *Zhang, Peng & Wang, 2018* |
| | | | | CNN + RNN | *Di Gangi, Bosco & Rizzo (2018)* |
| enhancer | one-hot-encoding | DNA sequence | human | CNN | *Yang et al. (2017)* |
| | | | | | *Min et al. (2016)* |
| promoter | embedding | DNA sequence | human | CNN | *Xu, Zhu & Huang, 2019* |
| | one-hot-encoding | DNA sequence | human | CNN | *Umarov & Solovyev (2017)* |
| | | | | | *Umarov et al. (2019)* |
| | | | human + *Oryza sativa* | CNN | *Pachganov et al. (2019)* |

| Table 2 (continued) | | | | | |
|---|---|---|---|---|---|
| Annotation | Preprocessing | Data | Species | Architecture | Reference |
| gene | one-hot-encoding | DNA sequence | metagenomics | CNN | *Al-Ajlan & El Allali (2019)* |
| pathogenicity | one-hot-encoding | DNA sequence | bacterias | CNN + RNN | *Bartoszewicz et al. (2020)* |
| species | embedding | DNA sequence | metagenomics | RNN + Attention | *Liang et al. (2020)* |
| | one-hot-encoding | DNA sequence | 13,838 species | CNN | *Busia et al. (2019)* |
| | | | viruses | CNN | *Ren et al. (2020)* |
| | | | | | *Tampuu et al. (2019)* |

## Annotation transfer

Annotation transfer relies on what the machine learning field calls "out-of-distribution generalization". The general idea behind is that when a neural network learns the link between a given annotation and a sequence, it may be able to generalize, that is to infer the annotation for another, albeit related sequence. While neural networks have been found to generalize well in this way (for a contemporary review, see *Shen et al. (2021)*), it is worth noting that it is always possible to design specific examples for which this generalization will fail (*Wiyatno et al., 2019*). For this reason, it is always a plus to have in hand some experimental data of the annotation you seek to predict. Even if the data has a low coverage, it can be used to validate or even fine tune the model using this new data (*Iman, Rasheed & Arabnia, 2022*).

The use of deep learning methodologies to annotate genomes from the sequence by transferring annotations learnt in a different context has been reported so far for three different applications. The first is the transfer from a species on which the network was trained to another species on which the predictions are made. The second one is the transfer of annotations from one cell type (or environmental condition) on which the network was trained to another cell type (or another environmental condition). This application relies on the use of secondary annotation that has to be used on top the DNA sequence to train the model.

### Annotation transfer across species

The potential of using deep learning to transfer annotations is especially relevant for different species since experimentally annotating all the sequenced genomes is today impossible. Here again, deep learning methodologies may help to close the gap by automatically annotating new genomes after being trained on reference, well annotated genomes. Initial studies provide a proof of principle of this possibility and highlight some of the limitations. *Khodabandelou, Routhier & Mozziconacci (2020)* demonstrate that a model trained on a given species can be used to annotate another species. They developed a model trained on the human genome to detect transcription start sites (TSS) and faithfully predict TSS on the mouse and on the chicken genomes with this model. Nevertheless, their model failed to generalise to other species such as the zebrafish. This study highlights some limitations of cross species prediction as this possibility relies on the conservation of molecular mechanisms through evolution. The conservation of

annotation logic has been further illustrated in *Minnoye et al. (2020)* and *Chen, Fish & Capra (2018)* in the context of enhancer prediction. *Minnoye et al. (2020)* studied the gene expression level associated to enhancers in melanoma for six different species. They demonstrated that the melanoma chromatin accessibility landscape is conserved for homologous enhancers and that the associated TF motifs are also conserved between the six species. More practically, *Chen, Fish & Capra (2018)* demonstrated that an enhancer predictor trained on one species among human, macaque, mouse, dog, cow and opossum performs correctly on all the other species. A model trained on species A and applied to species B has an AUROC score equal to 96% of the AUROC score of a model trained on B and applied to B (this figure is 85% for AUPRC). Taken together, these studies show the potential of training a model for a specific task on a given context and applying it to another, a process known as generalization. A promising idea in the field would be increasing the number of species on which the model is trained to increase its cross-species generalization. *Kelley (2020)* showed that deep learning models can be improved when trained on both the human and the mouse genome, especially in the context of predicting the effect of non-coding variant. This study shows that the diversity in the training set provided by training on two species helps the model generalise in the context of predicting the effect of mutations. Pushing the idea further, *Cochran et al. (2021)* showed the benefits of training on multiple genomes to increase the cross species generalization of models in the context of TFBS predictions. They also improved their predictions by developing a model which penalises the learning of species specific features during the training.

We expect cross species learning and prediction to play a major role in the near future in automatic annotation of ill-annotated genomes. In order to be successful these approaches will need to overcome two potential pitfalls. The first is that the further away species are on the tree of life, the less conserved are molecular mechanisms. The second being the potential over-fitting of a species specific logic. Overfitting is a modeling error in statistics that occurs when a prediction is closely aligned to the training set of data but fails to predict other unseen data. To avoid these pitfalls, testing the predictions with experimental cues is needed to confirm or not the computational predictions. The computational predictions are just one line of evidence about the true annotation. The strength of that evidence depends on how well the model has predicted out-of-distribution instances in the past and how different the distribution of interest is to the training distribution and to other tested distributions. The best solution in an unknown case would be to get a small dataset of experimental annotations to be used to validate and/or fine tune the models.

### Annotation transfer across cell type

In bio-medical research, a few cell types are considered as reference models due to their availability or their potential to be grown in culture. These cell types are extensively studied while for the overwhelming majority of cells types in the human body there is a blatant lack of data. Developing methods that could extrapolate the annotation of the reference cell types to the others can help address this lack. With this goal in mind, specific neural network approaches are developed to annotate the genome in a cellular context that

differs from the training context. Knowing that the DNA sequence is conserved between cell types, these methodologies use the DNA sequence and some complementary cell type specific annotation as input.

*Nair et al. (2019)* developed a CNN based model to predict the chromatin accessibility across human cellular context. The model uses the DNA sequence and the gene expression as input and predicts the local DNA accessibility. They evaluate the model genome wide on 10 cell types which were not used in the training set (113 cell types). The model achieves an average area under the precision recall curve (AUPRC) of 0.76 and an area under the receiver operating characteristic curve (AUROC) of 0.954 across five folds. *Quang & Xie (2019)* investigated the possibility of cross species prediction in the context of TFBS prediction. They developed a model to predict cell type-specific transcription factor binding from the DNA sequence, the gene expression, the DNaseI cleavage profile and the mappability. The model was also evaluated on cell types that differ from the ones used for training. On 51 TF/cell pairs on which the model was evaluated, it typically achieves an AUROC above 0.97 for most of the TF/cell pairs with a more contrasted figure regarding the AUPRC (between 0.21 and 0.87 depending on the pair). These two studies show promising results for the task of extrapolating annotations from reference cell types to other cell types.

A related problem is the prediction of the annotation of a genome in a different environmental context. *Liu et al. (2019)* showed that deep learning methods can be used to predict the gene expression in yeast from the DNA sequence and from 472 TF and signal molecules binding contexts for different stress conditions. Their evaluation was done on the same stress condition as their training as it is not possible to predict gene expression across different conditions. Their model was nevertheless able to perform in-silico TF knock-out experiments that were validated by micro-array experimental results.

## Unveiling sequence determinants of genomic annotations

We have seen above that when trained appropriately on large datasets, neural networks are capable of predicting annotations in new contexts. These neural networks in some ways mimic the machinery of DNA binding: metaphorically, they both "read" the DNA sequence and "annotate" it, with computationally predicted biochemical labels or molecules. Such a neural network can be treated as an emulation of the DNA binding machinery and used for *in silico* experiments to generate hypotheses and theories, like any other model. The real benefit comes from the ease to perform new "experiments" with this neural network and from the possibility to reveal how the model makes its predictions. In other words, the advantage of studying a biological mechanism with a neural network is twofold. First, by studying how the model makes its prediction, one can discover the motifs associated with the biological mechanism in the form of a PWM. Going further, a successful dissection of the model can give access to the grammar of motifs, *i.e.* understanding how motifs interact between themselves, forming motifs of motifs. Methods used to deal with these two tasks are described in the insert "Opening the black box". Second, a model can be use to predict the effect of non-coding mutations, also called variants, on the annotation. Variants are changes in the genomic sequence relative to

the reference sequence. They arise from the natural variability of individual DNA sequences and can be either single nucleotide variations (also called SNPs for Single Nucleotide Polymorphisms), or insertions or deletions of small sequences. The interpretation of the effect of variants within coding sequences is another topic that relates to genetics rather than genomics. In genetics, that covers protein functions and how they are affected by mutations, deep learning is also a game changer but this whole field will not be discussed here. Interested readers may refer to this recent work and reference therein for gene function predictions (*Brandes et al., 2022*) and to this comparative study for protein physical and chemical properties predictions (*Xu et al., 2020*). Non-coding variants can statistically be associated with phenotypic traits or diseases but their mechanistic role cannot be immediately inferred. The statistical approach, also know as Genome Wide Association Study (GWAS), reveals variants that are significantly over-represented in people with a certain trait. This analysis has an important drawback: many variants are linked, *i.e.* their co-occurrence is significantly more frequent, but within these linked variants, some have no role in creating the phenotype. In other words, GWAS is prone to false positives. Here again, deep learning can be used to prioritize variants, *i.e.* trying to find the variants responsible for the trait. In order to uncover how the DNA sequence drives the local assembly of various chromatin context we review below the different experimental datasets that have been studied using a deep learning based approach.

### Epigenomics: transcription factor binding site, histone modification and chromatin accessibility

The DeepSEA model (*Zhou & Troyanskaya, 2015*) was specifically developed to study the effect of non-coding SNPs on a huge set of epigenomic data (from 690 TF ChIP-seq, 125 DNase-seq and 104 histone marker ChIP-seq experiments). Despite the fact that the network was trained using a unique human reference genome, it is able to predict the decrease in DNase-seq sensitivity for 57,407 experimentally identified SNPs and these changes were confirmed by experiments. As a paradigmatic example, the network is able to predict the deleterious effect of SNP rs4784227 for FoxA1 protein binding, a mechanism associated with breast cancer. These encouraging results led the authors to use DeepSEA in a general way to discriminate variants associated with functional modification from innocuous variants. The network obtained at the time the best results on this task. Other teams have, since then, improved the quality of predictions on the same experimental dataset and same testset (*Tayara & Chong, 2019*; *Quang & Xie, 2016*).

The Basenji network (*Kelley et al., 2018*) was also used to predict the effect of variants. This network predicts the outcome of 2,307 ChIP-seq experiments on histone marks, 949 DNase-seq experiments, and 973 CAGE experiments in human. The predictions of the model in the presence of variants known to alter gene expression were compared to predictions obtained with non-variant sequence. Again, this network, although trained on a unique reference genome, is able to predict a change of the chromatin context at these loci. The model is for instance able to predict the effect of a variant (rs78461372) on the

two surrounding genes, one of which is located 13 kbp away. Again, predictions were confirmed experimentally.

*Liu et al. (2018)* demonstrated that a neural network could be used to identify mechanistically disease-associated SNPs from SNPs that co-occur with them. They developed a model to predict the chromatin accessibility given the DNA sequence. A test set was created consisting of 29 SNPs known to be related to breast cancer and 1,057 harmless SNPs that co-occur with them. A score quantifying the variations in network predictions were found to be significantly higher on disease-associated SNPs than on co-occurring SNPs (one-sided Mann-Whitney U test, $p\_value = 1.63 \times 10^{-3}$). The network training protocol can improve predictions associated with SNPs. *Hoffman et al. (2019)* used a CNN to predict the signals associated with one DNase-seq experiment and three ChIP-seq experiments on histone marks from the DNA sequence. They used the genomes of the individuals on which the experiments were performed as a source of sequences and not the reference genome. They defined a score to evaluate the consequences of 438 million variants. They showed that SNPs with a link to a disease or modifying the expression level of a gene are often attributed with a higher score. *Wesolowska-Andersen et al. (2020)* emphasize the importance of training the network on data obtained on a cell type which is relevant to the studied disease. In order to study the effect of variants associated with type II diabetes, they targeted islet cells of the pancreas. They developed a CNN predicting epigenomic data from the DNA sequence and show consistency between the network prediction-based method and traditional methods for refining the detection of diabetes-associated variants. A part (roughly 10%) of the initial set of variants can be labeled as important by looking at the model predictions. The authors show that those variants are indeed significantly more likely to be evolutionary conserved than the original set (one sided Wilcoxon rank sum test, $p\_value = 7.3 \times 10^{-4}$). Using this methodology, they were able to find 80% of expression quantitative loci (eQTL, loci associated with a quantitative trait) present in the variant set.

A number of biomedical studies have demonstrated practical application of deep learning for variant analyis and SNP interpretation. Illustrating this every day increasing number, a large study of congenital heart disease was recently performed (*Richter et al., 2020*). Another practical application of variant analysis using neural networks is provided by *Zhou et al. (2019)* to study variants related to autism. The study uses 7,097 genomes, 1,790 of which are from siblings, and overall covering 127,140 SNPs. These siblings' groups are formed by one member diagnosed with autism while the other is not. On average, the alleles of individuals with autism have a higher effect on transcription than the alleles of their siblings. The effect is larger when SNPs are close to genes and in particular close to loss-of-function intolerant genes Finally, 34 SNPs considered particularly important were experimentally tested. For 32 of them, a significant modification of the expression of associated genes was observed and among these genes many are active during brain development.

### RNA binding

Deep learning methodologies have also provided new insight into RNA binding processes. For example, by interpreting the first-layer filters (see insert "Opening the black box") of a CNN designed to predict RNA binding sites on the genome, *Wang et al. (2018a)* were able to identify new patterns of triple bond formation (between the two DNA strands and RNA). The authors also validated their finding with experiments. *Koo et al. (2018)* developed a CNN model to predict the binding sites of RNA-binding proteins. Analysis of their network by *in silico mutagenesis* (see insert "Opening the black box") showed that it was sensitive not only to consensus sites but also to their number and spacing. This analysis also revealed that the network learned to take into account the RNA secondary structure.

### DNA and RNA methylation

*Angermueller et al. (2017)* developed a neural network capable of predicting methylation sites from the DNA sequence. An analysis of the filters in the first layer associated with the observation of the predictions finds that GC-rich motifs tend to decrease the methylation of nearby CpG islands in contrast to AT-rich motifs. The motifs associated with the filters (see insert "opening the black box") were then compared with many TF consensus motifs. This analysis showed that 17 filters out of 128 correspond to TFs involved in methylation while 13 others are close to motifs of enzymes involved in methylation.

The miCLIP-seq protocol can be used to measure N6-methyladenosine (m6A) methylation on RNA. *Zhang & Hamada (2018)* used these data to train a network to detect methylation positions on mRNA sequences. By analyzing the filters in the first layer, they were able to find patterns associated with known m6A readers. Interestingly, they were also able to detect a reader of these methylations, *FMR1*, which was discovered *via* traditional methods in a paper published the previous year (*Edupuganti et al., 2017*).

### Gene expression

*Vaishnav et al. (2021)* trained a deep transformer network to predict the gene expression level associated with 20 million randomly sampled 80-bp long DNA sequences introduced in a *Saccharomyces cervisiae* promoter region. They assessed the effect of all single mutations in promoter regions and discovered four evolvability archetypes: robust promoters on which mutations have little effect, plastic promoters on which every mutations have a small effect and minimal or maximal promoters on which only some mutations can dramatically decrease or increase the associated expression level. Using this framework and analysing the promoter sequences in 1,001 yeast strains, the authors were able to demonstrate that evolution tends to select robust promoters. Earlier, *Liu et al. (2019)* trained a network to predict the expression level of yeast genes under different stresses. Analysis of the first convolution layer filters revealed that the network primarily searched for well-documented stress regulatory sites. Transcription factor silencing experiments *in silico* achieved results similar to real microarray experiments. *Zrimec et al. (2020)* also used a CNN to predict mRNA abundance directly from mRNA sequence in
*S. cerevisiae*. They demonstrated that the entire sequence is useful for determining the level of gene expression. Four elements (promoter, 5'UTR, 3'UTR and termination sequence) are used by the model to make the prediction. By interpreting the model with in silico *mutagenesis*, the authors recovered typical motifs of the four regions: TF binding motifs for the promoter or 5'UTR, the so-called Kozak sequence (*Kozak, 1989*) in the 5'UTR, poly-A and T-rich sites for the 3'UTR, and termination sites. More importantly, they demonstrated that mRNA abundance cannot be predicted by the presence or absence of the motifs alone, but can be predicted by the combination of motifs.

*Movva et al. (2019)* trained a network to predict the expression level of genes subjected to artificial regulatory sequences in humans. Interpretation of the network with DeepLIFT (see insert "Opening the black box") reveals that the sequences used by the network to make the prediction correspond to transcription factor binding sites. *Agarwal & Shendure (2020)* predicted gene activity from 10 kbp DNA sequences surrounding the TSS. The authors could not find motifs used by the network but an analysis of the over-represented k-mers in the promoters of highly active genes (according to the network) reveals the importance of CpG islands in predicting gene activity.

### Splicing, translation and polyadenylation of RNA

*Cheng et al. (2019, 2021)* developed a neural network to predict gene splice sites from the RNA sequence. Analysis of the effect of variants using this network shows its utility in understanding the genomic causes of autism. They used the dataset provided by *Zhou et al. (2019)* that we presented earlier and targeted 3,884 mutations that are near exons. They demonstrated that the disruption score of mutations as provided by their model is significantly higher in the affected group than in their unaffected siblings (Wilcoxom rank sum test, $p\_value$ = 0.0035). Once again the effect is larger in brain tissues. *Jaganathan et al. (2019)* confirmed the relevance of the use of neural networks for the study of gene splicing in the context of intellectual disability and autism. They used data coming from 4,293 individuals with intellectual disabilities, 3,953 individuals with autism spectrum disorder and 2,073 unaffected siblings. *De novo* mutations that are predicted to disrupt splicing are enriched 1.51-fold in intellectual disability and 1.30-fold in autism spectrum disorder compared to healthy controls.

Translation initiation of mRNAs does not always occur at the canonical AUG codon, as shown by the recent QTI-seq method which precisely maps translation initiation sites (*Gao et al., 2015*). These data have paved the way for the use of deep learning to predict these initiation sites. *Zhang et al. (2017)* developed a network capable of predicting initiation sites from mRNA sequences. By interpreting their network using input optimization methods, they highlighted the importance of Kozak sequences around AUG codons, confirming the previously established role of these sequences (*Kozak, 1989*).

Understanding the mechanisms controlling polyadenylation sites within mRNA sequences is another area that benefited from the contribution of deep learning methods. By interpreting their network, which is able to predict the probability of polyadenylation site usage in human mRNAs, *Leung, Delong & Frey (2018)* showed that poly(A) sites, the cutting factor UGUA, and GU-rich sequences tend to increase the probability of being

a polyadenylation site, whereas the presence of CA-rich sequences decreases this probability. *Gao et al. (2018)* also demonstrated the importance of poly(A) sites in the polyadenylation code in the plant *Arabidopsis thaliana* using a gradient-based method to interpret their network (see insert "Opening the black box").

### 3D genome structure

*Fudenberg, Kelley & Pollard (2020)* extended the Basenji method to predict the 3D structure of the genome directly from the DNA sequence. Targeted analysis of different areas of the human genome by in silico *mutagenesis* reveals that the CCCTC-binding factor (CTCF) binding sites are the most important elements for structure establishment. By testing the other TFs, the authors reveal that these have no influence apart from their possible interactions with CTCF. By performing CTCF site inversion experiments *in silico*, the authors show that the network is able to learn the role of CTCF motif direction in the 3D structure establishment. Finally, the attribution maps (see insert "Opening the black box") reveal the importance of cohesin ChIP-seq peak and, to a lesser extent, of promoters and enhancers.

### Nucleosome positioning

*Routhier et al. (2021)* used a CNN to predict nucleosome positioning in *S.cerevisiae* directly from the DNA sequence and evaluated the effect of every single mutation on the genome by in silico *mutagenesis*. They demonstrated the core role of the nucleosome depleted region (NDR) in nucleosome positioning and identified nucleosome repulsive motifs that were previously described in the literature. On the other hand, they did not find any motifs that would position nucleosomes by attracting them, suggesting that nucleosome repulsion is the main positioning mechanism. *Cakiroglu et al. (2021)* predicted nucleosome positioning as well as TFBS from the DNA sequence based on results obtained with Micrococcal Nuclease digestion treatment (MNase-seq, *Cakiroglu et al. (2021)*).
The model was able to reproduce the competition between nucleosomes and TFs for binding on the DNA. The analysis of the first layer of the CNN shows that the model identifies TF consensus motifs as important for the prediction and, by removing the filters corresponding to these motifs, the authors also demonstrated that TFs tend to exclude nucleosomes.

## Deep learning assisted genome writing

### Anticipation of experimental results and sequence fine tuning

Many cell or developmental biology experiments require the introduction of an artificial DNA fragment into the genome or modifying in some other ways the genome sequence. Having a neural network able to anticipate the consequences of these modifications on many genomic annotations allows to fine tune the experimental protocols not only by refining the sequence to introduce but also its position within the genome. The Kipoi repository gathers many independently developed networks. Its objective is to standardize and simplify the use of trained networks in concrete situations such as experiment support. For example, this repository makes available the DeepMEL network (*Minnoye et al., 2020*) developed to predict the accessibility of enhancers in melanoma in several

different vertebrate species. The effectiveness of this model to anticipate the expression of enhancer-associated genes has been demonstrated using the CAGI5 challenge data. DeepMEL can be used to predict the activity of artificially introduced enhancers and to optimize their sequence. In a related work in yeast, *Zrimec et al. (2020)* used their model to anticipate the genes expression level for various gene constructs, especially changing the terminator (5'UTR + termination sequence) with the promoters left intact. Their predictions were experimentally validated for six different genes and show great promise for the experimental control of gene expression by the sequence of surrounding regulatory elements.

The development of synthetic genomics is today largely due to the combination of the CRISPR-Cas9 protocol (*Jinek et al., 2012*), which allows to introduce tailored modifications in the DNA sequence in many organisms, with the industrialization of DNA synthesis (*Ostrov et al., 2019*). The CRISPR-Cas9 protocol uses small RNAs (sgRNA, single guide RNA) to guide the Cas9 protein to its target by sequence complementarity. However, sgRNAs usually target both the desired site and other sites on the DNA. Methodologies have been proposed to anticipate the binding strength between sgRNAs, the desired position and the spurious positions from their sequences (*Chuai et al., 2018*; *Xue et al., 2018*). These networks can be used to design sgRNA sequences that maximize interaction with the desired target and minimize interaction with spurious targets. In order to address the challenges of security and intellectual property raised by the development of synthetic genomics, Nielsen and Voigt have developed a deep learning model to predict the laboratory of origin of artificial plasmids from their DNA sequences (*Nielsen & Voigt, 2018*). For this specific question, however, deep learning methods do not necessarily deliver the best results (*Wang et al., 2021*).

### Synthetic sequence design

Possibly the most exciting prospective of the application of deep genomics is the computer assisted writing of genomes. Indeed, the use of neural networks to predict genomic functions from the sequence opens the possibility of optimizing sequences to control their function. This new research field has seen its first promising results in the recent years.

The study of transcription factor binding sites plays a key role in the application of deep learning to genomics, both for the development of architectures and interpretation methods. This problem has therefore naturally been approached from the perspective of sequence design. *Lanchantin et al. (2016)* optimized the input sequence to maximize the network predictions for specific TF binding. *Schreiber, Lu & Noble (2020)* developed Ledidi, a methodology to minimally modify the input sequence of the network in order to modify its predictions. Using this methodology, the authors are able to induce or destroy sites of CTCF binding or suppress JUND protein binding. *Gupta & Kundaje (2019)* used a method inspired by variational autoencoders to induce SPI1 protein binding sites.

On a different topic, *Bogard et al. (2019)* developed a sequence optimization methodology to design RNA sequences with controlled polyadenylation sites (*Bogard et al., 2019*). *Linder et al. (2020)* improved this technique to use it for various problems such as controlling the level of gene transcription, RNA splicing, or RNA 3' cleavage. More

recently, *Linder et al. (2021)* used masks on the sequence to both determine whether each part of the input sequence was sufficient to explain the network predictions and use this information to generate new sequences with similar properties. Other applications include *Cuperus et al. (2017)* who used their trained CNN to predict the translation level of mRNAs from their 5' untranslated sequence. This network was used to design 5' untranslated regions that induce maximal translation level.

*Vaishnav et al. (2021)* designed promoter regions that induced unusually low or high level of gene expression in yeast *S.cerevisiae*. They used a genetic algorithm to write an 80-bp long sequences that produce the desired output. The predictions were made with a deep transformer network trained to predict the gene expression level associated to 20 million randomly sampled promoters. Experimental validation on 500 sequences demonstrated that the sequences actually led to unusual level of expression. On average, designed sequences led to an expression level higher (or lower) than 99% of natural sequences. About 20% of the designed sequences led to a higher (or lower) expression level than any natural sequences.

## CONCLUSION

We highlighted in this review the high potential that deep learning holds to transform classical bioinformatics and open the deep genomics era. We started our tour by listing the first applications in the transfer of genomic annotations between species or between cell types. Due to the tremendous number of genomes that are sequenced everyday, we posit that deep learning will be a game changer for the task of genome annotation. We have also reviewed demonstrations of the potential of these techniques to uncover the complex regulatory grammar of motifs, which go beyond simple motifs represented by PWM that are of common use in the field of functional genomics. We finally presented perhaps the most transformative application of deep learning: the generation of new sequences using sequence optimisation based on predictions or using deep generative models.

Having reviewed these new potential avenues at the intersection between deep learning and genomics, we wish also to mention the risks that comes with using such techniques, as the output of an algorithm should always be taken with caution, especially in cases for which human health is at stake. In clinical medicine, initial enthusiasm for deep learning driven by over-stated results has given way to broad cynicism as the rubber has hit the road. See *Wynants et al. (2020)*, *Roberts et al. (2021)* for critical reviews of machine learning models for COVID-19 diagnosis.

A typical reason why predictions made by deep learning models may fail is out of distribution sampling. For instance, if the sequences on which a network is trained have an average GC content that is peaked around one value, there is no guarantee that predictions made on sequences harboring a different GC content will turn out to be correct. With that in mind, predictions that lead to annotation transfer, mechanisms discovery or sequence generation should be framed as a hypothesis generation tool to speed up research by suggesting targeted experiments.

We hope that our work will help colleagues in better understanding the profound impact that deep learning will have in the field of bioinformatics. We, as a community, now stand at the beginning of an exciting time: the deep genomics era.

## Sequence classification

### Species classification

Bioinformatics tools used to determine a short sequence's species of origin, such as BLAST, align and compare this sequence with sequences from different reference species. These tools are therefore increasingly slow as the number of reference species increases. Deep learning approaches do not suffer from the same problem. Indeed, once a network is trained to attribute a species to a DNA sequence, the prediction will always take the same amount of time. This advantage also comes at a cost: adding a new reference species to the database would require retraining the network. That retraining process is much more complicated, error-prone and demanding in terms of computational resources than adding a new sequence for BLAST. Networks have nevertheless already been developed for this specific application. The use of k-mer preprocessing allows the prediction of the species with a simple dense network (*Vervier, Mahé & Vert, 2018*). The results get even better when improving the sequence embedding strategy or increasing the length *k* of the k-mers (*Menegaux & Vert, 2019*, *2020*). Other methodologies use CNNs to predict the species of short ribosomal DNA fragments (*Busia et al., 2019*) or to identify viruses and microbes from metagenomic data (*Liang et al., 2020*; *Ren et al., 2020*). Another application that today shows great promise is the identification of viral DNA within metagenomic samples (*Tampuu et al., 2019*).

### Classifying and correcting sequencing errors

The variations of DNA sequences obtained from sequencers can be due either to the intrinsic diversity of the DNA sequence in the sample or to sequencing errors. In order to obtain the precise pool of sequences in a sample, it is necessary to differentiate the "true" variations from the sequencing errors. Several deep learning based methods have emerged for this purpose (*Poplin et al., 2018*; *Ravasio et al., 2018*). *Zhang, Shen & Huang (2019a)* proposed an improved method that leverages the internal states of a RNN to model the distributions of biased and unbiased RNA-seq reads. *Luo et al. (2018)* developed similar strategies for long read sequencing and *Torracinta et al. (2016)* developed a methodology that allows the correction of sequence errors for RNA-seq.

## Opening the black box

Neural networks are often referred to as "black boxes", which are trained to give the best answer but from which it is not possible to extract comprehensive rules. In the context of genomics, these rules would allows to understand which sequence motifs are associated with a given annotation. This chapter is intended at reviewing the methods used to "open the black box" in genomics studies and access the DNA motifs and their combination that are the most important for the prediction. Its content is more technical

than the rest of the review and it can be skipped by the readers who do not wish to get into these details.

We separate the different methods for interpreting deep networks into two categories. First-order methods allow to determine which DNA motifs play an important role in the network decision. Second-order methods allow to understand the grammar of motifs, *i.e.* a set of rules that is used to interpret motifs.

### First order–Motif discovery

Convolutional networks include filters (corresponding to short DNA motifs) that are optimized as they are trained. Within the first convolution layer, the network scans the sequence for the occurrence of motifs corresponding to the filters. Studying the motifs corresponding to the filters in the first layer is thus a way to see the DNA patterns deemed important by the network to make its predictions. *Kelley, Snoek & Rinn (2016)* and *Alipanahi et al. (2015)* introduced this method to study the Basset and DeepBind networks that they developed. Computing the motif associated with a filter requires three steps. First, all sequences in the test set are scanned using the filter. Second, for all sequences, at each position where the sequence matches the filter, a subsequence of the filter length is extracted. Matching here means that the norm of the elementwise multiplication between the subsequence and the filter exceeds a threshold. Third, the frequency of the nucleotides A, C, G, T within the extracted subsequences is computed to give a position weight matrix (PWM) of the motif searched by the filter. This method has a major disadvantage: there is no guarantee that the network searches for biologically important patterns with its first layer. This information can be dispersed within all layers. To overcome this limitation, constraints can be applied on the first layer of the network to make its weights directly interpretable in terms of a frequency matrix (*Ploenzke & Irizarry, 2018*). The network architecture can also be adapted to force information to be contained in the first layer (*Koo & Eddy, 2019*; *Koo & Ploenzke, 2020a*). An alternative option is to adjust the networks' training procedure to penalize the use of patterns that are too small and therefore not likely to be of biological interest (*Tseng, Shrikumar & Kundaje, 2020*). These methods work well for CNNs but cannot be used directly for RNNs. However, recurrent networks make intermediate predictions during their reading of the DNA sequence. Studying the positions that make these intermediate predictions vary the most can point at the nucleotides that are important in the establishment of the final prediction (*Lanchantin et al., 2017*).

A second class of method assigns to each nucleotide of a sequence a score reflecting its importance in the prediction made by the network. If the network predicts several classes, a score will be calculated for each class. There are two ways to compute this score. The first method was introduced for the study of DeepBind and DeepSEA. For each nucleotide, the difference between the predictions obtained with the natural sequence and with a mutated sequence is computed. By summing up the contributions of the three possible mutations a mutation score is obtained (*Alipanahi et al., 2015*; *Nair, Shrikumar & Kundaje, 2020*). This method is called in silico *mutagenesis*. The second method is based on the estimation of the change in the prediction $P_c(X_0)$, obtained for the class c, that

is obtained when the input, one-hot encoded, sequence $X_0$ is change to another sequence $X$. When $X$ close to $X_0$ the Taylor expansion to first order gives:

$$P_c(X) - P_c(X_0) \approx \left.\frac{\partial P_c}{\partial X}\right|_{X_0} (X - X_0) \tag{1}$$

The quantity $\left.\frac{\partial P_c}{\partial X}\right|_{X_0}$ allows to estimate in which proportion the change of an input element will affect the prediction. As a bonus, this quantity can be easily and cheaply computed with the same methods used during network training. Multiplying this quantity term by term with the input $X_0$ produces an importance score for each nucleotide of the input sequence (*Lanchantin et al., 2017*). Based on similar principles, the DeepLIFT (*Shrikumar, Greenside & Kundaje, 2017a*; *Ancona et al., 2017*) and DeepSHAP (*Lundberg & Lee, 2017*) methods also use back-propagation to compute how much a change in the input changes the prediction. Both are inspired from methods used in the image recognition field. Some of their potential limitations have been put forward in this field (*Sturmfels, Lundberg & Lee, 2020*) and it remains to be seen how these limitations will impact the interpretation of deep learning models in the context of genomics.

The importance scores assigned to nucleotides can be used to determine important motifs. *Avsec et al. (2021b)* developed a methodology, called TF-Modisco, to determine globally important motifs from the assignment scores. This methodology works in three steps. In the first step, an importance score is associated to all positions of the test set sequences with the DeepLIFT model. In a second step, all sub-sequences with high scores are extracted. In order to define these high scores, the scores of the real sequences are compared to the scores obtained for random sequences having the same di-nucleotide distribution. Finally, the sub-sequences are grouped into motifs using hierarchical clustering. Peaks in the importance scores can also be interpreted as peak of ChIP-seq data and standard bioinformatic tools such as MEME (*Bailey et al., 2015*; *Bailey et al., 2009*) can be used to extract important motifs (*Routhier et al., 2021*).

### Second order–Grammar of motifs

The methods used to explore the grammar of motifs can also be divided in two categories: methodologies that exploit the model architecture to compute the interactions between motifs and methodologies that take benefit of the attribution map to visualize the effect of varying the motifs organisation.

CNNs typically have multiple convolution layers. The filters of the second layer can be analyzed in the same way as the filters of the first layer and provide interactions between filters (*i.e.* motifs) of the first layer. Networks using an attention mechanism to exploit the patterns in the sequence are directly interpretable. The first convolution layer transforms the one-hot encoded sequence in which letters are replace by vectors of ones and zeros into a 2D matrix, one dimension representing the 1D sequence and the second dimension representing the different filters of the first layer. The attention mechanism assigns a weight to each point of the sequence, a weight learned during training. The "encoded" sequence is then averaged along the spatial axis, weighted by the attention

weights. Thus, spatial interactions between filters are readable in the weights assigned by the mechanism (*Hu et al., 2019*).

Another approach to decipher the grammar of motif is to perform *in silico* experiments consisting in introducing, moving or destroying motifs and assessing the impact on the predictions. *Greenside et al. (2018)* propose to study the evolution of a motif importance score upon mutation of another motif to discover possible interactions between them. *Koo & Ploenzke (2020b)* developed a methodology to quantify the global importance of any motif in a general context by evaluating the difference between the average of the predictions obtained for sequences randomly drawn from the natural distribution and the average of the predictions obtained for these same sequences in which the motif has been artificially included. This method can be used to analyze the interactions between motifs by adding two motifs within the random sequences. *Avsec et al. (2021b)* also performed *in silico* experiments to understand the grammar of motifs by changing the genomic distance between the motifs and assessing the evolution of the prediction.

## ACKNOWLEDGEMENTS

We would like to thank Lou Duron and Alex Westbrook for their comments on the manuscript. We are also grateful to the reviewers for their invaluable work.

### Funding
The authors received no funding for this work.

### Competing Interests
The authors declare that they have no competing interests.

### Author Contributions
- Etienne Routhier conceived and designed the experiments, performed the experiments, analyzed the data, prepared figures and/or tables, authored or reviewed drafts of the article, and approved the final draft.
- Julien Mozziconacci conceived and designed the experiments, performed the experiments, analyzed the data, authored or reviewed drafts of the article, and approved the final draft.

### Data Availability
This article is a literature review; there is no raw data.

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
