# Peer review of "Genomics enters the deep learning era"

_PeerJ, doi:10.7717/peerj.13613_

## Round 0.1 · original submission · Major Revisions

The manuscript appears to require major revision as it appears very abstract in areas and perhaps not as in-depth as would be needed to discuss biology-based applications. This does seem more like a rush to publication on a popular topic and closely appears to mimic other publications, even in a similar title style out in the wild. Granted that this was designed for an audience of biologists; however, larger depth in application areas might be also considered. One reviewer, considered expert in this field, identified many areas where the manuscript missed the mark. Though a review, the authors might benefit by including personal experience and results. I would recommend that the authors review comments carefully and reconsider submitting another submission which more clearly defines this space for biologists. I have seen similar reviews and would suggest a novel approach angle that may provide increased impact. The manuscript is being returned with suggestion for major revision.

·

Basic reporting

Authors provide a very good overview of deep learning architectures for functional annotation of genomes, prediction of the effect of non-coding mutations and generation of synthetic genomic sequences. The manuscript is clearly written and easy to follow.

Experimental design

The areas of interest, three in total, are clear and the selected publications are representative for these areas. As a literature review paper, the methods are adequate.

Validity of the findings

Major issues:
The conclusions section is limited. What is the main outcome of the manuscript? What can we learn from the selected publication ? Which are the most common DL architectures used and why ?
I would propose to the authors to provide also a discussion section where the limitations of deep learning approached in genomics are presented and a commentary of the low adoption of DL models in the field compared to other computer science fields such as vision or NLP.
A couple of sentences in the abstract about the discussion/conclusions of the review must be added.

Additional comments

Minor/Typos:
91 “evaluated on cell types that different”  “evaluated on cell types that are different”

Reviewer 2 ·

Excellent Review

This review has been rated excellent by staff (in the top 15% of reviews)
EDITOR COMMENT
From other engagements within the subject area under review I was concerned that the manuscript was not reaching far enough and sought feedback from known experts in the field. The feedback was on point and validated some concerns that the initial manuscript be re-evaluated to determine whether it should be pursued further or strengthened in areas of weakness. The reviewer in this case was very familiar with the subject area and capable of providing frank feedback on what would be expected in a review-type manuscript.

Basic reporting

The authors survey recent developments in the intersection of deep learning and genomics. As stated clearly in the introduction, the review is primarily for the benefit of biologists working in genomics and adjacent fields. The emphasis is on applications and results, rather than architectures and techniques, which distinguishes this review from others and makes it fit for PeerJ's scope.

They broadly group the contributions into three categories: annotation transfer, biological mechanism discovery, and synthetic genomics.

In annotation transfer, neural networks trained on experimentally-derived annotations in one context (environment, cell type, species) are used to generate annotations in another context (another environment, cell type, or species). For example, the authors discuss together the results of Khodabandelou et al. (2020), wherein transcription start site inference transfers across species, and the results of Quang and Xie (2019), wherein transcription factor binding site inference transfers across cell types.

In biological mechanism discovery, neural networks are probed (e.g. in their weights or in their predictions in response to perturbed inputs) and the outcomes interpreted in biological terms. For example, the authors share the results of Wang et al. (2018), wherein the learned weights in the first layer of a CNN trained to predict RNA binding sites were used to hypothesize a novel mechanism, later validated experimentally, for triple bond formation between double-stranded DNA and single-stranded RNA.

In synthetic genomics, neural networks are used to predict the features of entirely synthetic nucleotide sequences. For example, the authors share the results of Chuai et al. (2018) and Xue et al. (2018), which both use neural networks to optimize the single guide RNA sequences used to target gene editing using the CRISPR-CAS9 system.

This systematic review and its conceptual framework will no doubt be of great help to genomics researchers who want to navigate the difficult literature at the intersection of deep learning and genomics. As a deep learning researcher originally trained in biology, I benefited from exposure to several exciting new avenues of research at the juncture of two of my favorite fields.

However, there are multiple major issues that need to be addressed before the review is ready for publication.


Issue #1:

I find that the positioning and language in the review substantively misrepresent the strength and implications of the results in annotation transfer and in mechanism discovery. Both applications rely on the usage of neural networks in ways that are far from guaranteed to work. This is not obvious just by reading the review, which risks misleading non-ML experts into weighting evidence from deep learning models too highly, relative to other forms of evidence. In clinical medicine, initial enthusiasm for deep learning driven by over-stated results has given way to broad cynicism as the rubber has hit the road. See, e.g., Wynants et al. 2020 (doi: 10.1136/bmj.m1328) and Roberts et al. 2021 (doi: 0.1038/s42256-021-00307-0) for critical reviews of machine learning models for COVID-19 diagnosis.

Annotation transfer relies on what the machine learning field calls “out-of-distribution generalization”. While neural networks have been found to generalize in this way (for a contemporary review, see Shen et al. 2021, doi: 10.48550/arXiv.2108.13624), they also often fail, even catastrophically so, even in the easier case of in-distribution generalization (see adversarial examples: Wiyatno et al. 2019, doi: 10.48550/arXiv.1911.05268; and test set resampling: Recht et al. 2018, doi: 10.48550/arXiv.1806.00451), especially in the case of supervised methods, like most of those considered here. Unlike for classical machine learning methods, like linear models, theoretical or engineering guarantees on generalization behavior of neural networks have been elusive (classic result in Zhang et al. 2017, doi: 10.48550/arXiv.1611.03530). Even the problem of determining when a model is failing to generalize on closely-related data is challenging (for some empirical results see Rabanser et al. 2019, doi: 10.48550/arXiv.1810.11953; for recent work on limits to performance, see Ginart et al. 2022, doi: 10.48550/arXiv.1611.03530).

The review should make it clear that any annotations by a network with no exposure to the target domain are of a highly tentative nature – more akin to the results of a GWAS than of a mechanistic study (to say nothing of a pre-registered study or a body of literature). Further, they should note the common solution to this problem: fine-tuning models. Fine-tuned models are first trained on a closely related domain with rich data and then trained again on the target distributions where data is limited (for review see Iman et al. 2022, doi: 10.48550/arXiv.2201.09679). For example, one could train a model in the richly annotated context of HeLa cells and then, given a much smaller dataset of annotations in a data poor context, like cells in a newly-discovered species, fine-tune the model to give reasonable annotations in the new domain. Importantly, this means that the model performance in the target domain can be assessed on held-out test data from that domain. For more detail, see the “Validity of the Findings” section.

The results in biological mechanism discovery share these issues and add more. That is because mechanism discovery relies in some cases on out-of-distribution generalization, e.g. when the network’s inputs are perturbed and the predictions interpreted in biological terms. because this is a form of out-of-distribution data, the same caveats from annotation transfer apply. In these cases, mechanism discovery should be framed as a hypothesis generation tool to speed up research by suggesting targeted experiments (again, much like a GWAS).

Other forms of biological mechanism discovery involve “opening the black box” of the network and trying to learn, as scientists, what the model has learned. The interpretability methods used to obtain some of these results (gradient-based attributions, Shapley values) are very controversial in the deep learning community. Contrary references must be included in the “Opening the black box” insert alongside the explanations of the methods and a hint of the unsettledness of this science should appear in the text. For more details, see the “Validity of the Findings” section.

Note that the use of neural networks in synthetic genomics has the same issues, but the use cases presented in the review take this into account. For example, neural networks are positioned as a method for generating candidate CRISPR guide RNAs, with experimental verification.

Issue #2:

I noticed a large number of small factual errors in the text, described in the “Validity of the Findings” section. They include mathematical, computational, and conceptual errors. Because I am an expert only in the deep learning half of deep learning for genomics, I can only comment on the errors there, but they do not inspire confidence in the material in which I am not an expert.

I urge the authors to carefully review all their claims about presented results, not just the ones identified in my review, before resubmission.


Issue #3:

Finally, there are a number of issues of language (spelling, grammar, and comprehensibility) in the manuscript and of type-setting and organization in the tables. The tables in particular are a key contribution of the paper, as they help interested readers navigate the literature quickly at a high level, so their organization and formatting is important.

I have identified these issues in detail in the “Additional comments” section.

Experimental design

The methodology, based on searching PubMed and similar online databases, is simple but effective. There are no quantitative claims about the literature in the text, so there is no need for a more sophisticated review strategy.

The organization of the literature into three broad classes based on application is clean, clear, and of great utility to researchers interested in new approaches to long-standing genomics problems.

Validity of the findings

My most substantive concerns with the manuscript, on which I am basing my request for major revisions, regard the validity and factual correctness of the content.

As described in the “Basic Reporting” section, the largest issues concern overstating the results based on out-of-distribution generalization and model interpretation. I will here pick out specific areas where I think .

On line 96, the authors mention that Liu et al. (2019) test in “the same stress condition as their training” and note that this “fail[s] to firmly demonstrate” out-of-distribution generalization. The language here should be much stronger. There is no way, from the given results, to evaluate whether the model in question generalizes to other stress conditions. I would suggest moving this result to the top of the annotation transfer section and pointing out this methodological issue and the ways in which it is avoided (and not avoided) in other results. For example, Minnoye et al. (2020) have strong evidence for a model that generalizes across species but don’t provide evidence for generalization across cell types, since all samples are from melanoma cell.

On lines 130-132, the authors say “experimental cues may be needed in some case [sic] to confirm or not [sic] the computational predictions”. This presents external validation of out-of-distribution results as the exception, rather than the rule. Instead, it should be stated that the computational predictions are just one line of evidence about the true annotation. The strength of that evidence depends on how well the model has predicted out-of-distribution instances in the past and how different the distribution of interest is to the training distribution and to other tested distributions. The prospect of a small dataset of experimental annotations being used to quickly transfer models should be explicitly discussed, since that is where deep learning experts and genomics experts can collaborate most fruitfully to expand the scope of annotated genomes.

On lines 134-136, the authors say “In a metaphoric [sic] way, neural networks are trained to emulate the mechanistic link between the DNA molecule and the cell nucleus machinery, including TFs, that "read" the DNA sequence. The biological complexity is replaced by the complexity of the neural network.” This is a beautiful and potentially useful metaphor but it dangerously overstates the alignment between what neural networks learn and biological fact. It gives, along with the claims in the following section, the impression to non-experts that neural network predictions can be trusted as though they were the outcomes of biological experiment

A better way to make a similar rhetorical move without the overstatement might be to say: “We’ve seen above that when trained appropriately on large datasets, neural networks are capable of predicting annotations in new contexts. These neural networks in some ways mimic the machinery of DNA binding: metaphorically, both “read” the DNA sequence and “annotate” it, with labels or molecules. Such a neural network can be treated as a rich model of that DNA binding machinery and used for in silico experiments to generate hypotheses and theories, like any other model.”

On line 290, the authors claim that one of their studies on nucleosome positioning “demonstrated that attractive motifs are unlikely to exist”. In reviewing the paper (doi: 10.1101/gr.264416.120), I only found a much weaker claim: ”we do not find any motifs that would position nucleosomes by attracting them”. As I am not an expert on the genomics here, I would appreciate a clarification of how the much stronger biological claim is supported.

As mentioned in the “Basic reporting” section, there has been much research into the shortcomings of the attribution methods used for some of the mechanism discovery results. For concerns about the objectivity of Shapley-style methods, see Sturmfels et al. 2020, doi: 10.23915/distill.00022, and the excellent references therein. In particular, Adebayo et al. 2018 (doi: 10.48550/arXiv.1810.03292) demonstrate that some saliency methods can report the same results even when weights are randomized or when the model is trained with random labels.


There are also a number of places where results from other papers are described in purely qualitative terms (“is relevant”, “performs correctly”) rather than quantitative terms (“achieves accuracy five percentage points higher than a BLAST baseline”, “reduces the number of false positives by a factor of 10, relative to traditional methods”, “achieves comparable accuracy with only”). These kinds of quantitative comparisons are more common for evaluating models in the deep learning literature and should also be more immediately interpretable to an expert genomics audience.

Some examples of qualitative results are listed below, with line numbers.

87. “predictions are relevant”
91. “studies shows [sic] promising results”
115. “performs correctly on all the other species”
183. “variations in network predictions were found to be significantly higher”
194. “show the consistency between”
258. “show its utility in understanding”
259. “confirmed the relevance”
354. “led to unusual level of expression”.

Expanding on line 354, which is regarding Vaishnav et al. (2021)’s transformer-validated, synthetically evolved promoter sequences: were unusual levels the desired outcome? How is “unusual” quantified? Why is it not “desired” levels of expression? Are there benchmark comparisons, e.g. to human-designed promoters?

Some smaller points:


On line 228, the authors indicate that Zhang and Hamada (2018) identified a “new” m6A reader “FMR1, which was unknown to date”. This is incorrect. Zhang and Hamada indicate that FMR1 was determined to be an m6A reader via traditional methods in a paper published the previous year: Edupuganti et al. (2017), doi: 10.1038/nsmb.3462.

On line 282, the authors reference Zeng et al. (2018)’s “NLP-inspired” model. In that paper, they used a shallow, Word2Vec-style embedding layer and gradient-boosted treess. These are NLP-flavored methods but they are not deep learning methods, because there is no stacking of differentiable modules. This paper and its results should be removed from the review.

In the “Sequence classification” insert, the following claim appears: “Indeed, once a network is trained to attribute a species to a DNA sequence, the prediction will always take the same amount of time.” While this is true, it is a somewhat unfair comparison. BLAST gets better at classification with each added reference species, while a neural network only gets better when it is retrained. That retraining process is much more complicated and error-prone than adding a new sequence for BLAST. For large neural networks, retraining can also take several days.

Regarding the “Opening the black box” insert:
The equation in this section is incorrect. The left-hand side is an exact quantity and the right-hand side is the first-order term of the Taylor expansion for approximating that quantity. These two quantities are not in general equal. Either the higher-order terms must be represented (using Landau notation, adding + O(X - X0)) or, since this is a less technical audience, the equation should be replaced with an approximate equals sign (e.g. \approx in LaTeX).

Additionally, it is not entirely correct that the derivative of the class with respect to the input is computed during network training. During training, we calculate the derivative of the loss with respect to the parameters (typically, weights and biases of linear layers). It would be more correct to say “this quantity can be easily and cheaply computed with the same methods used during network training”.

Additional comments

The grouping of research is great and a wonderful contribution, but the second section, which I’ll call “biological mechanism discovery”, after its identifier in the tables, is somewhat confusingly presented. In the abstract (line 13), it’s “prediction of the effect of non-coding mutations”. In the end of the introduction (lines 53-54), it’s “getting a deeper understanding of the molecular mechanisms”. In Figure 2’s caption (above line 61) it’s “get mechanistic insights on a process involving the DNA sequence”. The title of the relevant section (line 133) is “Unveiling sequence determinants of genomic annotations”. This makes it a bit harder to see how the authors have split the literature cleanly into groups. I recommend unifying the language (e.g. by consistently using the phrase “biological mechanism” in each place).

What an ML researcher would call a “network” or “model” is here referred to as a “methodology”. If this is standard in the world of deep genomics, that doesn’t have to change, but even then it would be better to align with the broader world of ML.

143. Both inserts are labeled “insert 3”. There are only two inserts.
148. “covers protein functions and will not be discussed here”. A brief comment on the difference between genomics and genetics and a citation for applications of deep learning to the latter would be appropriate here.
197. “study uses 7097 genomes including 1790 siblings”. Are all of the genomes from siblings or just some? The following description leaves this unclear to me.
215. “learned by itself” is excessive anthropomorphization of the ML algorithm. Suggest deleting “by itself”.
222-223. “17 filters” and “13 others” needs more context. How many filters are there total? “17 filters out of X” would be sufficient

“filters” (line 210) and “in silico mutagenesis” (line 214) are used without definition, but they are defined in the “Opening the black box” insert. Readers should be directed to the insert for the definition.

The following terms are not defined:
129. “over-fitting”
130. “generalisability” (which should be generalisation)
304. “pre-trained”.
In “Black box” insert: “regularization” and “cost function”. I would suggest instead “An alternative option is to adjust the networks’ training procedure to penalize the use of patterns that are too small”, which avoids introducing the two technical terms.
In “Black box” insert: “one-hot encoded”. If the authors think that this term is known among genomics researchers, it need not be defined, but it is italicized, so perhaps they intended to define it.


The manuscript might be improved by the following citations:

46. For CNNs in computer vision: the Neocognitron (Fukushima 1980, doi: 10.1007/BF00344251) and LeNet (LeCun et al. 1989, doi: 10.1162/neco.1989.1.4.541).
266. A citation for the role of Kozak sequences around AUG codons
273. “the gradient method” – which? Add relevant citation. Should also direct readers to the relevant insert.
281. “attribution maps” – which? Add a citation. Again, direct readers to the relevant insert.
315. A citation for CRISPR.


Suggestions for improvements to the tables:

The architecture column has inconsistent levels of detail, e.g. in “self-attentive gated convolutional highway” for Guo et al. (2021) versus “CNN” for many others. I would change it to “Architectural Features” and pick them from a small list: “convolutions”, “recurrence”, and “attention” is enough.
The “preprocesing” column seems unnecesssary. Almost all methods use a one-hot encoding method, especially in Table 1. If it is critical to include, could add “embeddings” to the list of “Architectural Features”.
The second page of Table 1 is split into two pieces, but it’s unclear why.
Table 2 and Table 3 have the same columns and title. Why are they separately numbered tables?
The spacing and placement of the words is distractingly poor, e.g. where Guo et al. (2021)’s “Annotation” column entry crosses over
The capitalization and spellings of many of the words are inconsistent: “Syn.” vs “syn”, “genomic” vs “genomics”, “Tranfer” in place of “Transfer”


Below are detailed corrections to the grammar, spelling, and language in order of line number. To be clear, these are suggestions, and if the authors disagree on any, they are welcome not to incorporate that particular change and there is no need for a detailed response. However, the manuscript is generally in need of copy and prose editing to improve clarity.

24. “publication” should be “publications”
36. Missing right parenthesis after “(2018)”
43. “Among the first and emblematic” should be “most emblematic”
55-56. “classification methodologies such as machine learning” is incorrect. ML encompasses more than just classification. Suggest “from” in place of “such as”
73. “second in the transfer” should be “second is the transfer”
74. “network was trained and another species” should be “network was trained to another species”
79. “can help filling this lack” is incorrect. Suggest “can help address this lack”
85. “across human cellular context” should perhaps end in “contexts”
87. “inverstigated” should be “investigated”
89. “cell type specific” should be “cell type-specific”
91. “cell types that different from” should be “cell types that differ from” or “cell types different from”
91. “two studies shows” should be “two studies show”
95. “472 TF and signal molecules binding context” should perhaps end with “binding contexts”
110. “enhancers prediction” should be “enhancer prediction”
118. “”generalisability” should be “generalisation”
119. “non coding variant” should be “non-coding variants”
128-129. Sentence “The first is the lack of conservation of molecular mechanisms at work between species that are too far away on the tree of life.” is awkwardly constructed. Suggest “The first is that the further away species are on the tree of life, the less conserved are molecular mechanisms.” or similar
131. “experimental cues may be needed in some case” should end in “cases”
141. “i.e” should be “i.e.”
142-143. “described in the insert 3” should be “described in Insert 3”
146. “Single Nucleotide Polymorphism” should be “Single Nucleotide Polymorphisms”
147. “within the coding sequences” should be “within coding sequences”
149-150. In “their mechanistic role is often difficult to comprehend.”, “comprehend” is a strange word choice. Suggest rewriting “a mechanistic role cannot be immediately inferred.”
150. “also know as Genome Wide Association Studies” should be “also known as a Genome Wide Association Study”. The appositive refers to “the statistical approach”, so the plural is not correct.
151. “These analysis” should be “These analyses”
153. “GWAS are” should be “GWAS is”. While it is true that GWAS is sometimes used as a plural, in this context the text is referring to the statistical approach, singular.
156. “data-sets” should be “datasets”
163. “,it” is missing a space after the comma
169. “data-sets” should be “datasets” and “test-set” should be “test set”
182. “with the former” could be shortened to “with them"
191. Missing year in citation of Wesolowska et al.
194. “show the consistency” should perhaps be “show consistency”
196. “A practical application of variant analysis using neural networks is provided by Zhou et al. (Zhou et al. (2019)) to study variants related to autism.” This intro line should be re-written. See comment below on lines 204-207.
199. “the other are” should be “the others are” or “the other is”, depending on whether the sibling groups are always pairs or are sometimes larger.
199. “Studying siblings allows to compare the effect of SNPs in genetically very close individuals” is grammatically incorrect. Suggest removing, because the role of sibling studies should be clear to anyone with interest in genomics
201. “The difference is even reinforced when” should be “The effect is larger when”
204-207. The sentence starting “Illustrating the” is something of a non-sequitir, since the rest of the paragraph is organized around relating a different result. Suggest starting the paragraph with “A number of biomedical studies have demonstrated practical application of variant analyis and SNP interpretation.”, then quickly explaining the Richter et al. (2020) results, then explaining the Zhou et al. (2019) results.
208. Section heading ends with punctuation, unlike other headings
210. When you first introduce filters, point to “Opening the black box” insert for definition.
212. “sites were experimentally validated” is a mis-use of passive voice. It should be clear whether the experimental validation was done by the same group that made the prediction, even if it was a posteriori
218. “from the sequence” should identify which sequence (DNA? RNA? something else?). From context it is clear that it’s DNA, but there are lots of sequences. Note that the phrase “DNA sequence” is used e.g. on line 286 but this is not done consistently.
229. Section heading ends with punctuation, unlike other headings
231. “associated with 20 millions” should end with “20 million”
231. “80-bp long sequence” should be “80-bp long sequences”
231. “s.cerevisiae” should be capitalized. Typical style is to use the whole Linnaean name the first time: “Saccharomyces cervisiae”.
232. “promoters regions” should be “promoter regions”
234. “mutation have” should be “mutations have”
243. “5’UTR” uses a different mark here than elsewhere
245. “poly-A” is written differently here than elsewhere. Should perhaps define as “polyadenylated” in this first use
250. “see the insert 3” should be “see Insert 3”
252. “kpb” should be “kbp”
257. “from the sequence”: which? See comment on line 218
261. comma at the end of “canonical AUG codon”
263. deep learning is italicized here and in several other places when it should not be
265. Kozak sequences are referred to as “so-called” here but not in their first introduction. Suggest only using “so-called” at the first introduction or not at all
268. deep learning s italicized here and in several other places when it should not be
273. “Arabidopsis Thaliana” should be “Arabidopsis thaliana”
277. “CTCF” is not defined here, which is the first usage
282. “NLP” should be “natural language processing” at first usage
284. “the loop forming activity” should be just “loop-forming activity”
286. “the nucleosome positioning” should just be ”nucleosome positioning”
291. “the nucleosome positioning” should just be ”nucleosome positioning”
291 “from the DNA sequence, more precisely” should be “from the DNA sequence. More precisely,”
292. “the result of the MNase-seq protocol considering that” is an awkward construction. What precisely are they predicting and how exactly is it related to nucleosome positioning and TFBS?
294. “competition between nucleosome and TF” should be either “the nucleosome and TFs” or “nucleosomes and TFs” – I believe the latter
302. “but also its position” occurs without an accompanying “not only” or “not just”. Unclear from context what should go here, because “refining the sequence to introduce” is an incomplete fragment.
309. “perspective” does not seem to be the right word here. Is it “related work” or “a related approach”?
312. “great promises” should be “great promise”
314 “the combination of the CRISPR-Cas9 protocol” is incorrect – if you’re not referring to separate protocols, there’s nothing to combine. I believe “the CRISPR-Cas9 protocol” is correct here
321-323. Instead of putting the issue of IP and security first, put the Nielsen et al. (2018) reference first, then explain why it’s important.
324. “from sequence” should be “from their sequences” or “from their DNA sequences”. SSee comment on line 218
325. Is Nielsen et al. on line 323 the work cited here as “Nielsen and Voigt (2018)”? These should both be Nielsen and Voigt.
325. deep learning is italicized here and in several other places when it should not be
328. “exciting perspective of the application” is incorrect. I believe it should be “exciting prospective application”
332. deep learning is italicized here and in several other places when it should not be
333. There is an extra right parenthesis after “deep learning”
336. “developed Ledidi a methodology” should be “developed Ledidi, a methodology”
341. “Going further these applications on TFBS” is ungrammatical
345. “to alternately determine” should be “to both determine”
347. “MRPA-trained CNN” is the first appearance of MRPA, so the acronym should be defined
351. “s.cerevisiae” should be capitalized to “S. cerevisiae”
351. “to evolve 80-bp long sequence toward a sequence that” should perhaps be “to produce an 80-bp long sequence that”. This is shorter and avoids confusion with in vitro evolution
352. “a sequence that induce” should be “a sequence that induces”
353. “associated to 20 millions” should end with “20 million”
354. “The” is unneeded at the beginning of the sentence
358. “bioinformatic” should be “bioinformatics”
359. “between cell type” should be “between cell types”
359. “amounts of genomes” should be “number of genomes”
361. “We have also shown the potential” seems incorrect. That phrasing is usually used to refer to reported experimental findings. Instead, perhaps “We have also reviewed demonstrations of the potential”
363. “the most transformative application of deep learning, which is the generation” could be shortened to “the most transformative application of deep learning: the generation”
366. “will and already has” is ungrammatical.
366. “bioinformatic” should be “bioinformatics”
366-367. “apologies to the colleagues whose work was not referenced here, even if we did our best to cover this vast and rapidly evolving field.” is not a great ending. First, if this is truly a systematic review, there’s not much to apologize for. Second, it’s tonally quite different from the rest of the manuscript. Finally, it’s a weak foot on which to end. I would suggest removing this sentence and ending with a statement about the excitement and possibilities of the “deep genomics era”.

Note: there are no line numbers in the insert, so the remaining edits are not placed on a particular line but instead numbered in order of appearance.

“Sequence classification” insert

“approaches should allow to overcome this problem.” should be “approaches do not suffer from the same problem.”
Last sentence in Species classification section: “great promises” should be “great promise”
“Luo et al. developped” should be “Luo et al. developed”


“Opening the black box” insert


“coined as” is incorrect. Change to “referred to as” or “considered”
The definition “The interpretation of neural networks is defined here as a set of analysis that reveals, at least partially what are the important features that are recognized by the network to make a correct prediction.” feels out of place, since the term “interpretation” isn’t used very much and the definition isn’t particularly precise.
“a set of analysis” should be “a set of analyses”
“at least partially” should end in a comma
“genomic networks” is confusing – it could refer to networks of interactions at the genome level. Change to “deep networks in genomics” or just “deep networks”, since these methods are all general.
In the final paragraph in the first section of this inset, two sentences are fragments: the one beginning “First-order methods, that” and the one beginning “Second-order methods, that”. Removing “, that” would fix this issue.
“elementary multiplication” should be “elementwise multiplication”
“patterns with its first layer this information can be” should be split into two sentences: “patterns with its first layer. This information can be”
“dispersed within all layers by successive combinations” is unclear. I suggest removing “by successive combinations”.
“To overcome this criticism” feels incorrect. It is not a criticism but instead a “limitation” of neural networks.
“CNN” and “RNN” should be pluralized to “CNNs” and “RNNs”
“second class of methods is to assign” should be “a second class of methods assigns”
in silico mutagenesis is finally defined here after being used in the text multiple times. At its first usage, point the reader to this insert.
I was unable to understand the explanation of gradient-based methods: “The second method is based on the estimation of the change in the prediction Pc(X0), obtained for the class c, that would be obtained in the input sequence changes from X0 as input to X.” What does it mean to have an “input” to X, which is not a function?
“gives the equation 1:” is incorrect. First, the definite article is not needed. Second, there is only one equation in the paper, so there’s no need to call out which it is. Third, the colon is used to direct the reader to the equation, so a number is superfluous. I suggest “gives:” and removing the annotating (1) from the equation.
“allows to obtain” is incorrect. Change to “produces”
The last two sentences of the first paragraph of the “Second order – Grammar of motifs” section are fragments: the one beginning “First methodologies, that” and the one beginning “Second. methodologies, that”. Removing “, that” would fix this issue.
“the attention mechanism” is incorrect, as there are multiple types of attention mechanism. Change to “an”.
“one-hot encoded” is italicized and should not be
Last paragraph: “An other” should be “another”
“the grammar of motif, is” should be “the grammar of motifs is”
“in which the motif artificially included” should be “in which the motif has been artificially included”.

---

## Round 0.2 · accepted · Accept

The manuscript appears more focused and reads well. With the popularity of AlphaFold2 being used to predict protein structures I would have expected some sort of lead-in for the next phase of this work as protein structure can be predicted, and deep learning might be applied to identify genome variation affecting such genome diversity from a reference. However, I did like the treatments taken on different stages of nucleic acid architecture and for the applications which would be applied. I did not see a multitude of specifics for the machine learning tools; however, there is just enough with references which may lead interested parties into a more detailed treatment of the data. This manuscript should be a good starting-point introduction for biologists to dive-in deeper into genome analysis. I will approve the manuscript to be moved forward from this stage. Congratulations.